# Systematic multi-trait AAV capsid engineering for efficient gene delivery

Fatma-Elzahraa Eid [1,2] ✉, Albert T. Chen [1], Ken Y. Chan[1], Qin Huang[1], Qingxia Zheng[1], Isabelle G. Tobey [1], Simon Pacouret[1], Pamela P. Brauer[1], Casey Keyes[1], Megan Powell[1], Jencilin Johnston[1], Binhui Zhao[1], Kasper Lage [1,3,4,5], Alice F. Tarantal[6], Yujia A. Chan [1] & Benjamin E. Deverman[1] ✉

Broadening gene therapy applications requires manufacturable vectors that efficiently transduce target cells in humans and preclinical models. Conventional selections of adeno-associated virus (AAV) capsid libraries are inefficient at searching the vast sequence space for the small fraction of vectors possessing multiple traits essential for clinical translation. Here, we present Fit4-Function, a generalizable machine learning (ML) approach for systematically engineering multi-trait AAV capsids. By leveraging a capsid library that uniformly samples the manufacturable sequence space, reproducible screening data are generated to train accurate sequence-to-function models. Combining six models, we designed a multi-trait (liver-targeted, manufacturable) capsid library and validated 88% of library variants on all six predetermined criteria. Furthermore, the models, trained only on mouse in vivo and human in vitro Fit4Function data, accurately predicted AAV capsid variant biodistribution in macaque. Top candidates exhibited production yields comparable to AAV9, efficient murine liver transduction, up to 1000-fold greater human hepatocyte transduction, and increased enrichment relative to AAV9 in a screen for liver transduction in macaques. The Fit4Function strategy ultimately makes it possible to predict cross-species traits of peptide-modified AAV capsids and is a critical step toward assembling an ML atlas that predicts AAV capsid performance across dozens of traits.

Engineering novel functions into proteins while retaining desired traits is a key challenge for developers of viral vectors, antibodies, and inhibitors of medical and industrial value[1–3]. For instance, to be harnessed as a viable gene therapy vector, an adeno-associated virus (AAV) capsid should simultaneously exhibit high production yield and efficiently target the cell type(s) relevant to a specific disease across preclinical models to human patients. A common approach for developing AAV capsids with novel tropisms is to funnel a random library of peptide-modified capsids through multiple rounds of selection to identify a few top-performing candidates. This approach has

[1]Stanley Center for Psychiatric Research, Broad Institute of MIT and Harvard, Cambridge, MA, USA. [2]Department of Systems and Computer Engineering, Faculty of Engineering, Al-Azhar University, Cairo, Egypt. [3]Department of Surgery, Massachusetts General Hospital, Boston, MA, USA. [4]Novo Nordisk Foundation Center for Genomic Mechanisms of Disease, Broad Institute of MIT and Harvard, Cambridge, MA, USA. [5]Institute of Biological Psychiatry, Mental Health Center St. Hans, Mental Health Services, Copenhagen, Denmark. [6]Departments of Pediatrics and Cell Biology and Human Anatomy, School of Medicine, and California National Primate Research Center, University of California, Davis, CA, USA. ✉e-mail: fatma@broadinstitute.org; bdeverma@broadinstitute.org

produced modified capsids that more efficiently transduce cells throughout the central nervous system (CNS)[1,4–8], photoreceptors[3], brain endothelial cells[9,10], and skeletal muscle[11,12]. These rare capsids can then be diversified to screen for even more enhanced tropisms[4], high production yield[4], or cross-species functionality[12]. However, variants optimized for one trait may be difficult to co-optimize for other traits, and the protein sequence space is too vast to effectively sample by chance for rare variants that are enhanced across multiple traits. As a result, AAV engineering teams often devote many years and significant resources to developing capsids that ultimately fail to be optimized across multiple traits essential for preclinical and clinical translation.

To identify novel and diverse AAV capsids that simultaneously possess multiple traits relevant to gene delivery (e.g., manufacturability, targeting to disease-relevant cells across host species, detargeting from other cell types), vast capsid sequence spaces must be subjected to systematic and unbiased searches. This quickly becomes intractable using traditional methods as the capsid sequence is increasingly modified. For instance, merely inserting a string of seven amino acids (a 7-mer) into an AAV9 capsid generates a theoretical sequence space of 1.28 billion variants; inserting a 10-mer instead of a 7-mer extends that space to 10 trillion variants.

To address this challenge, we sought to develop a generalizable ML-guided approach to systematically and simultaneously map 7-mer-modified AAV9 capsid sequences to multiple traits of interest. To generate high-quality data that would enable the training of accurate ML models, it was necessary to first create a low-bias, high-diversity library composed only of "production-fit" capsid variants that are capable of assembling and packaging a genome (Fig. 1). This "Fit4-Function" library was subjected to in vivo and in vitro screens for traits relevant to gene therapy, which, as anticipated, resulted in highly reproducible data that could be used to train robust ML models. The models trained using the Fit4Function data were of sufficient accuracy that they could be leveraged in combination to search the much larger, untested, theoretical production-fit sequence space in silico for rare multi-trait variants. We first demonstrated that six models relating to production fitness and liver-targeting in vitro and in vivo could be used in combination to predict sequences that met filters set across all six

models. The resulting library of variants exhibited a high 88.4% validation rate, i.e., 88.4% of its variants were experimentally determined to fulfill all six criteria. Despite being trained only on mouse in vivo and human in vitro data, this combination of six models translated to the macaque. Variants nominated for individual validation performed well across human cells and mice compared to AAV9. These same variants provided more efficient transduction of the macaque liver relative to AAV9 when tested in a pooled library format. Notably, the combination of in vivo and in vitro functional predictors boosted the precision of cross-species prediction compared to the use of any individual model. In other words, although it has been argued that human cell models are less useful than animal models in AAV gene delivery vector development, we observed that models trained on data from human cell in vitro functional assays were valuable for predicting variants that exhibit the trait of interest in mice and macaques in vivo. The Fit4-Function approach allowed us to systematically identify the combination of traits that is most critical in predicting a given function of interest; appropriate screening models can be identified and used to enrich for multi-trait capsids. This strategy can inform intelligent searches for AAV capsids that are performant across species and more likely to translate from preclinical models to investigational human gene therapies.

## Results

### Production fitness sequence space mapping

We and others have successfully derived enhanced gene delivery vectors from AAV9 capsids modified through the insertion of seven amino acids (7-mer) between VP1 residues 588–589. To create an accurate and generalizable sequence-to-production-fitness ML model, synthetic modeling and assessment libraries were designed for the purposes of training and validating the model, respectively. Each consists of 74.5K variants that uniformly sample the sequence space (each amino acid was sampled with an equal probability at each position); 10K of the 74.5K are common to both libraries to assess reproducibility across libraries. This is distinct from conventional NNN or NNK (where N is any base and K is a G or T) libraries where millions of variants are synthesized stochastically by uniformly sampling the nucleotide space, which biases toward amino acids represented by

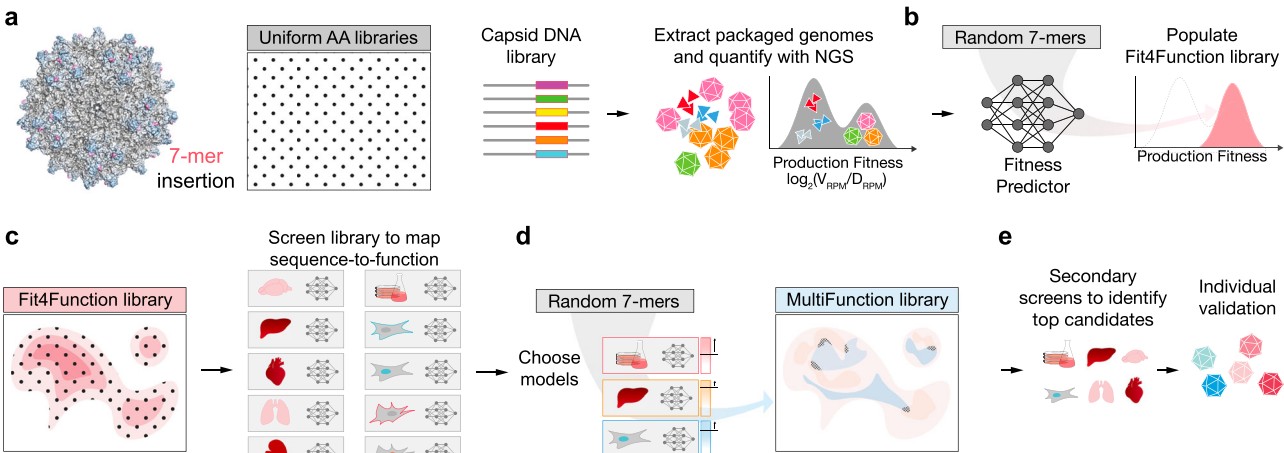

**Fig. 1 | Systematic multi-trait protein optimization paradigm. a** An insertion-modified AAV library that uniformly samples the 7-mer sequence space (1.28 billion possible variants) is designed and used to produce AAV particles. Variant production fitness is measured via Next-Generation Sequencing (NGS) of nuclease-resistant *Cap*-containing genomes ($V_{RPM}$) relative to the number of genomes in the DNA library ($D_{RPM}$). **b** The production fitness data is used to train a sequence-to-production-fitness ML model that is then used to design the Fit4Function library, which uniformly and exclusively samples the production-fit sequence space. **c** The

Fit4Function library can be screened in vitro or in vivo for functions of interest, and the data are used to derive ML models that predict these functions from random 7-mer sequences. **d** The production fitness and functional fitness models are used in combination to populate MultiFunction libraries consisting of variants predicted to perform well across the desired traits (see checkered areas that represent the overlap between the functional sequence spaces of interest). **e** The MultiFunction AAV libraries are produced and screened for all functions of interest. The top-performing variants are then individually validated.

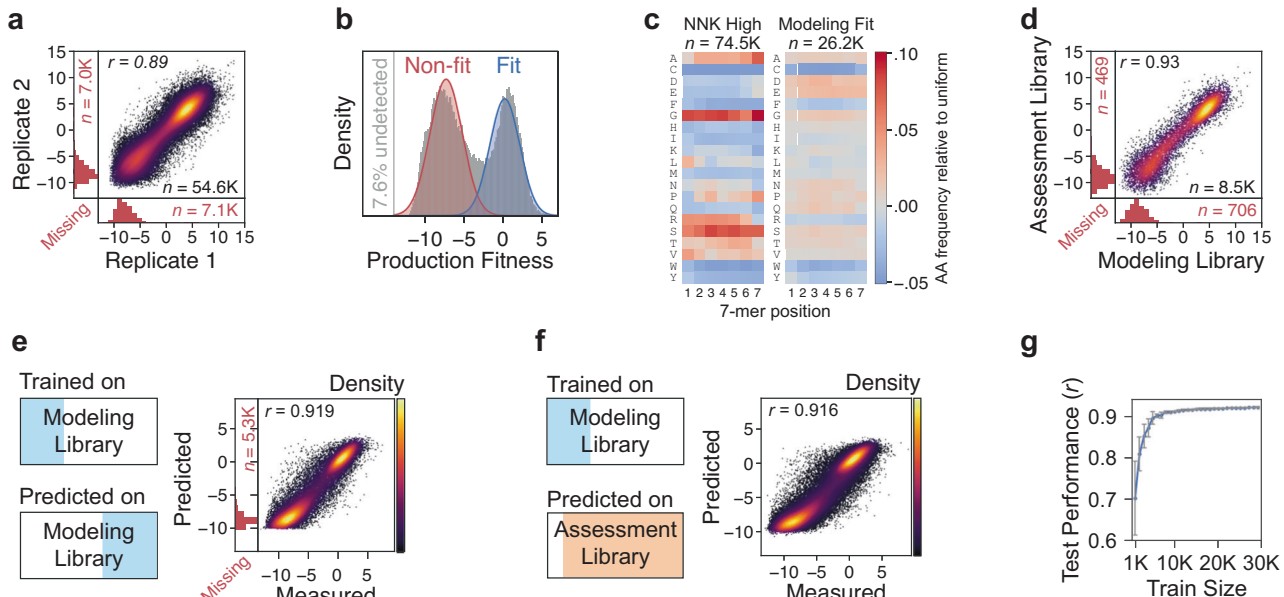

**Fig. 2 | Mapping and learning the 7-mer production fitness landscape. a** The correlation between the production fitness scores of codon replicate pairs is shown. The vertical and horizontal marginal histograms correspond to missing cases where only one codon replicate of a pair was detected. **b** The production fitness distribution of the modeling library represents the variants detected in at least one of the 24 replicates (92.4% of total variants). The distributions representing non-fit versus production-fit variants are depicted. **c** The amino acid distribution by position for the variants in the 74.5K most abundant sequences in an NNK library versus the production-fit distribution of the modeling library (26.2K out of 74.5K). **d** The production fitness replication quality is shown for the control set (10K) that is shared between the modeling and assessment libraries. The Pearson correlations between the predicted versus measured production fitness scores are shown when the model is trained on a subset of the modeling library and **e** tested on another subset of the same modeling library ($n = 30.6K$) versus when **f** tested on the independent assessment library, not including the overlapping 10K set ($n = 57.7K$ after removing the undetected variants). **g** The performance of the production fitness prediction model is shown across different training set sizes ($n = 10$ models, mean ± s.d.). Source data are provided in a Source Data file.

more codons. Both modeling and assessment libraries were also designed to assess whether codon usage impacts production fitness; each variant is represented by two maximally different nucleotide sequences (codon replicates, i.e., replicates that encode the same seven amino acid residues but using different codons to serve as in-sample biological replicates). We produced both libraries in triplicate, in two separate runs, by two different researchers, for a total of twelve replicates each. The reproducibility (measured by the agreement between replicates) of variant production fitness scores between preparations by different researchers improved as technical, codon, and biological replicates were aggregated (Supplementary Fig. 1). Therefore, we performed all subsequent analyses on production fitness scores aggregated across all replicates for each library.

We first assessed whether codon usage impacts the production fitness of identical amino acid variants. If so, it would be necessary to train on the nucleotide sequence space ($61^7$ for NNN, $31^7$ for NNK), which is much larger than the amino acid sequence space ($20^7$). We observed a high correlation between the production fitness scores of the codon replicates in the modeling library (Fig. 2a and Supplementary Fig. 2), suggesting no significant codon usage bias. Therefore, we averaged production fitness across codon replicates for all downstream modeling.

The production fitness distribution of the modeling library could be modeled by a mixture of two Gaussian distributions: a "non-fit" versus a "production-fit" distribution (Fig. 2b). Note that the "production-fit" distribution includes variants that produce better than, as well as, or less well than AAV9; "production-fit" is not defined as having a production fitness score greater than that of AAV9. The non-fit distribution overlaps with the production fitness distribution of the stop-codon-containing variants, which are presumably detected at low levels in the AAV library due to cross-packaging (Supplementary Fig. 3). The variants in the production-fit distribution exhibit distinguishing amino acid sequence characteristics such as a general

enrichment of negatively charged residues and the depletion of cysteine and tryptophan (Fig. 2c). Nonetheless, this production-fit distribution had less bias than an analogous set of the most abundant 74.5K variants from an NNK library (Fig. 2c). The production fitness scores for the 10K variants common to both libraries were consistent (Fig. 2d), suggesting that variant fitness is not noticeably impacted by the other variants in each of the libraries.

## A generalizable production fitness model

While prior studies applied classification models to predict AAV capsid production fitness[13,14], we used a regression model to capture the large variation in relative production fitness scores (±5-fold, $\log_2$ enrichment) within the production-fit and non-fit distributions. We first trained the model using the sequence and production fitness measurements of 24K variants unique to the modeling library. The accuracy of each model in this study was assessed by the agreement (Pearson correlation) between the measured fitness scores and the model's predicted scores. Remarkably, the sequence-to-production-fitness model achieved high accuracy on the remaining subset of the modeling library not used in the training process (Fig. 2e), as well as the independent assessment library (Fig. 2f). In addition, the model does not require large amounts of training data to obtain high accuracy, reducing the training from 24K to 5K variants only slightly reduced performance ($r = 0.924 ± 0.001$ vs $r = 0.899 ± 0.015$, Fig. 2g). These data demonstrate that the model is generalizable across libraries and to unseen variants and requires relatively small training datasets given the high quality of the data.

## Fit4Function enables reproducible data and accurate prediction models

Using the production fitness model, we randomly generated and predicted the fitness of 24M variants in silico. The predicted production-fit sequence space was then uniformly sampled for 240K variants to

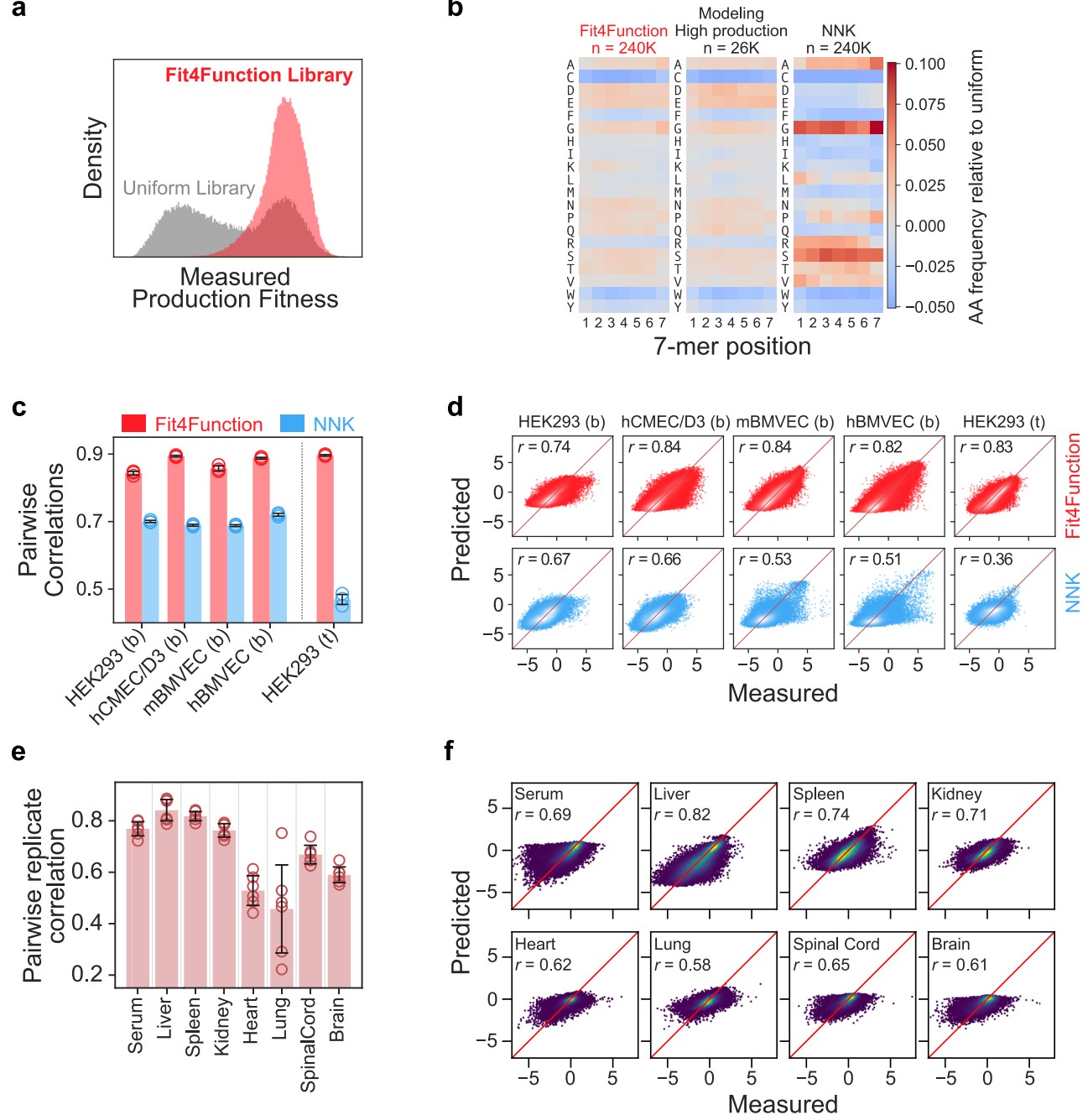

**Fig. 3 | Fit4Function libraries uniformly sample the production-fit space and enable more accurate functional screening and prediction. a** The distributions of the measured fitness scores are shown for 100K randomly sampled variants from the Fit4Function library versus the uniform modeling library. **b** The amino acid distribution by position for the variants in the Fit4Function library, production-fit distribution of the modeling library, and 240K most abundant sequences in an NNK library are shown. **c** The pairwise Pearson correlations among biological triplicates across functional screens (mean ± s.d.; one-tailed paired $t$-test, $n = 5$ screens; $p = 0.0065$) using the Fit4Function library (240K) versus an NNK library (top 240K variants) are shown. hCMEC/D3: human brain endothelial cell line, mBMVEC:

primary mouse brain microvascular endothelial cells, hBMVEC: primary human brain microvascular endothelial cells, HEK293: HEK293T/17 cells. Binding and transduction are indicated by 'b' and 't', respectively. **d** The measured versus predicted functional fitness ($\log_2$ enrichment scores) for models trained on Fit4-Function versus NNK library data are shown. **e** The replication quality (mean ± s.d.) between pairs of animals ($n = 6$ pairs across four animals) for the Fit4Function library biodistribution in eight tissues is shown. **f** The prediction performance of models trained on the in vivo biodistribution of the Fit4Function library across eight organs is shown. Source data are provided as a Source Data file.

create a "Fit4Function" library. As expected, the measured fitness scores for the Fit4Function variants, when synthesized, mapped to a single distribution that closely follows the production-fit distribution of the modeling library (Fig. 3a). The amino acid distribution in the Fit4Function library is similar to that of the production-fit distribution from the modeling library and is similarly less biased when compared

to that of the 240K most abundant variants in an NNK library (Fig. 3b). It is important to note that the production-fit threshold used in populating Fit4Function libraries is not so stringent as to eliminate potentially promising functional candidates for downstream optimization; only non-fit variants with poor production (those whose production fitness is comparable to stop-codon-containing control

sequences) are not sampled for Fit4Function libraries. Nonetheless, the Fit4Function approach enables users to set their own fitness thresholds, e.g., one could require that the majority of the capsid variants in their library produce at least as well as AAV9.

Fit4Function libraries are designed to enable the generation of reproducible and ML-compatible functional screening data. Specifically, the library is limited to a moderate size that enables deeper sequencing depth and samples only production-fit variants, which both enable more quantitative and reliable detection of each variant in the library. In addition, the library samples the production-fit amino acid sequence space in an even manner, which results in less biased and more generalizable ML models.

To demonstrate the high-quality, ML-compatible screening data enabled by the Fit4Function approach, we first compared the outcomes of our screening strategy using the Fit4Function library versus an NNK library across five functional assays: (1) HEK293 cell binding, (2) primary mouse brain microvascular endothelial cell (BMVEC) binding, (3) primary human BMVEC binding, (4) human brain endothelial cell line (hCMEC/D3) binding, and (5) HEK293 transduction. Binding and transduction were measured by quantitative sequencing capsid variant abundance at the DNA and mRNA levels, respectively. The Fit4Function library consistently yielded higher replication quality data than the NNK library (one-tailed paired $t$-test, $n = 5$ assays; $p = 0.0065$; Fig. 3c).

Second, we built and compared models trained on functional data derived from the Fit4Function library versus an NNK library (only data from the most abundant 240K variants in the NNK AAV library were used). The Fit4Function-based models consistently achieved higher prediction accuracy (Fig. 3d). These results are unsurprising because NNK libraries have theoretically millions of random variants, some of which can be at levels that are two or more orders of magnitude higher than other variants in the library. As a result, state-of-the-art sequencing is not able to accurately quantify a great number of the under-represented variants in NNK libraries before and after screens. In addition, the unknown variant membership in NNK libraries and the resulting low-quality data mean that ML models cannot be trained to discriminate between truly non-functional variants and variants that cannot be reliably detected. In comparison, Fit4Function libraries that are sized at 240K variants or less have a manageable, known membership and focus on production-fit variants, which increases the likelihood of detection for each variant. Critically, this allows for accurate functional learning and predictions from both positive and negative data. Furthermore, Fit4Function libraries used for training functional fitness models are populated with variants sampled uniformly across the sequence space. Compared to NNK libraries, the higher replication quality and reduced bias in Fit4Function libraries enable more accurate modeling.

We next sought to examine the use of the Fit4Function library to train prediction models for more complex functions where NNK screening data would typically be sparse and challenging to use for training accurate models. Here we considered in vivo AAV biodistribution after systemic administration in four adult C57BL/6J mice. The replication quality was high in the liver, kidney, and spleen, and moderate in the brain, spinal cord, heart, lungs, and in serum (Fig. 3e and Supplementary Fig. 4). We trained independent models to predict the variant tropism for each organ. The training data measurements were aggregated across three animals, and the data from the fourth animal was held out for independent testing. The models performed reasonably well when trained on assays with more reproducible data (Fig. 3f; model performance correlated with the data replication quality shown in Fig. 3e), demonstrating the applicability of our approach to in vivo data.

## Multi-trait capsid identification

Efficient and durable gene delivery to the liver remains challenging due to vector efficiency, capsid antigen presentation, and T cell-mediated immunity. Liver-directed therapies should benefit from the development of more potent AAV vectors that can be administered at lower doses to reduce exposure to capsid antigens. Previous efforts to develop capsids with improved human hepatocyte transduction have generated candidates that are selective for human hepatocytes but inefficiently transduce the mouse liver in vivo[2,15]. While such vectors have important translational potential—one capsid, LK03 is now being evaluated in human clinical trials (NCT05092685; NCT04581785; NCT03003533; NCT03876301)[2]—there is a need for capsids that exhibit improved tropisms across species so that they are also compatible with preclinical efficacy and safety testing.

Our objective was to design a MultiFunction library consisting only of variants that are each predicted to possess multiple enhanced functions related to cross-species hepatocyte gene delivery. Notably, AAV9 has a strong liver transduction profile in animals compared to other AAV serotypes, including AAV5 and AAV8[16]. Therefore, an AAV9-based capsid engineered to efficiently target the liver across species could be a candidate for a liver gene therapy vector. We performed five separate functional screens of the Fit4Function library: (1) binding or (2) transduction of the human hepatocellular carcinoma cell line (HEPG2), (3) binding or (4) transduction of the human liver epithelial cell line (THLE-2), and (5) efficient liver biodistribution in C57BL/6J mice (Supplementary Fig. 5). We used the high-quality data from these functional screens to train and assess the performance of five independent sequence-to-function models (Fig. 4a). With the production fitness model and these five functional fitness models, we screened 10M randomly generated variants in silico and selected 30K liver-targeted MultiFunction variants predicted to have enhanced phenotypes across all five functions and production fitness (an "enhanced phenotype" was arbitrarily defined as any variant above the 50th percentile of the measured enrichment scores). In the MultiFunction library, each variant was encoded by two nucleotide sequences serving as codon replicates. In addition, we included 3K variants from the modeling library (production-fit and non-fit; Uniform Control), 10K from the Fit4Function library (Fit4Function Control), and 3K from the known hits in the Fit4Function library, i.e., variants from the Fit4Function library that had been experimentally confirmed to exhibit enhanced phenotypes for the five hepatocyte-related traits and production fitness (Positive Control).

To assess the accuracy of our multi-trait predictions and identify the top-performing variants, we screened the MultiFunction library using the same five assays related to hepatocyte targeting and for production fitness (Supplementary Fig. 6a–c). The MultiFunction variants either matched or surpassed the performance of the positive controls from the Fit4Function library (Fig. 4b); 88.4% of the MultiFunction library variants satisfied our enhanced phenotype definition as compared to 2.6% of sequences in the uniform space or 7.0% of the Fit4Function (production-fit) space (Fig. 4c). Although the 7-mer sequences in the MultiFunction library have an increased frequency of arginines and lysines, the library diversity remains high (Supplementary Fig. 6d).

We individually assessed the performance of seven variants that were selected from the MultiFunction library based on their measured production fitness, liver biodistribution and transduction in mice, and their enhanced ability to bind and transduce human HEPG2 and THLE-2 cells (Fig. 4d). Each capsid and AAV9, as a control, were used to package a single-stranded green fluorescent protein (GFP) and Luciferase dual reporter AAV2 genome. Production yields were comparable to that of AAV9 (Supplementary Fig. 7a). When administered to mice at $1 \times 10^{10}$ vector genomes (vg)/mouse and assessed for GFP expression three weeks later, each capsid and AAV9 efficiently transduced hepatocytes as assessed by the native GFP fluorescence in DAPI[+] liver nuclei (Supplementary Figs. 7b, c and 8). Each individually validated MultiFunction variant was more effective (10–1000-fold) than AAV9 at transducing the HEPG2 and THLE-2 cell lines (Fig. 4e and Supplementary Fig. 7d).

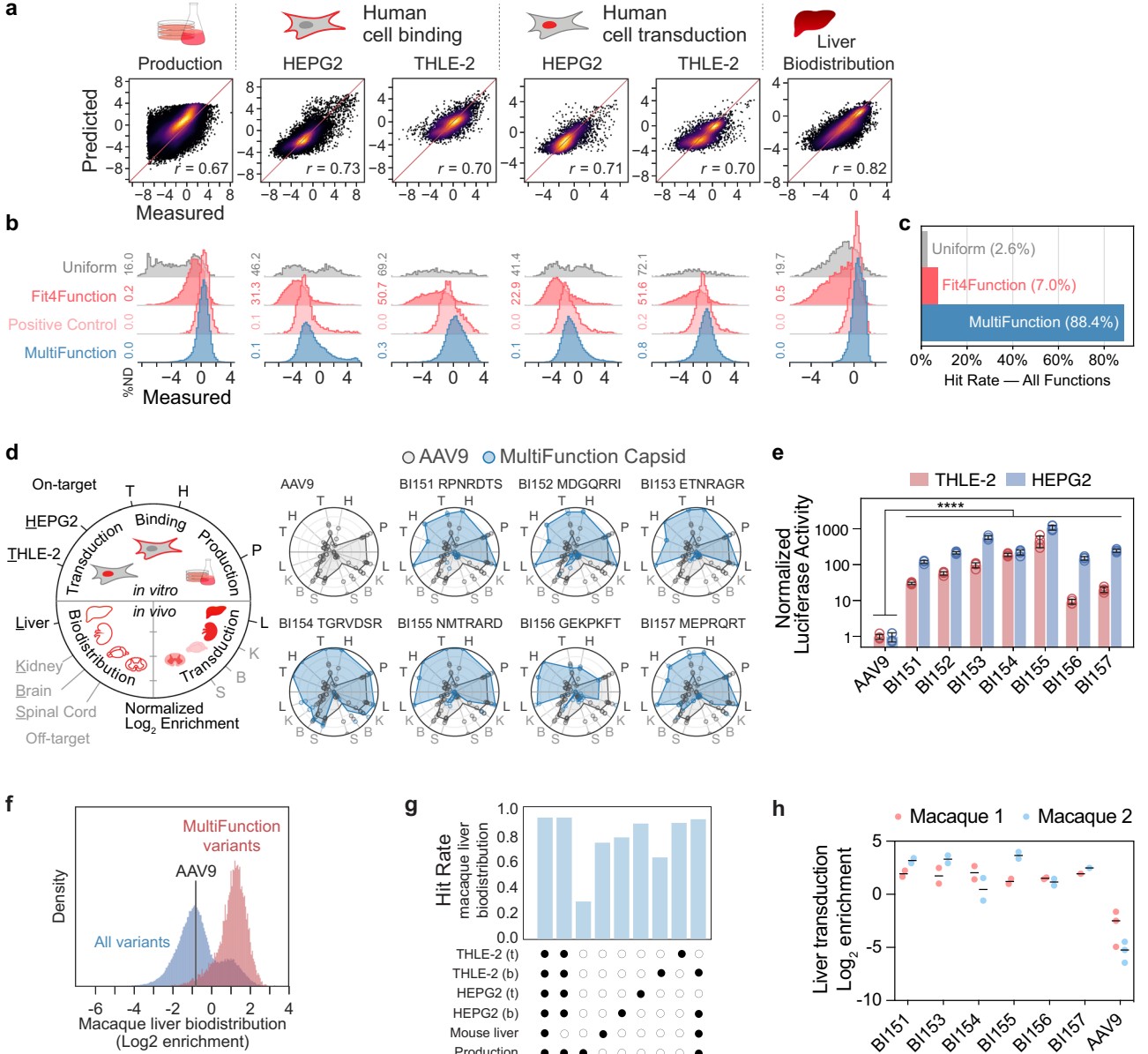

**Fig. 4 | Liver-targeted MultiFunction library design, validation in human cells and mice, and translation to macaque. a** The Pearson correlation of measured versus predicted enrichment are shown for production fitness and hepatocyte-targeting assays. **b** The enrichment distributions are shown across variants sampled from the Uniform, Fit4Function, Positive Control (Fit4Function variants satisfying the six conditions), and MultiFunction libraries. The histograms are density-normalized, including non-detected variants (ND). **c** The hit rates are shown for variants satisfying the six conditions in each listed variant set. **d** The on-target and off-target measurements for capsids BI151–157 and AAV9 in the MultiFunction library pool are shown as log₂ enrichments of the selected capsid (two codon replicates) as compared to AAV9 (four codon replicates). The measured enrichment was linearly normalized according to the maximum and minimum values for each assay. Individual replicates are plotted as points. The average normalized enrichments across replicates are plotted as polygon vertices. **e** HEPG2 and THLE-2 transduction were assessed 24 h post-transduction at 3000 vg/cell using a luciferase assay (*n* = 4 transduction replicates per group, mean ± s.d., ****p < 1e−4, unpaired, one-sided *t*-tests on log-transformed values, and Bonferroni corrected for multiple hypotheses). The measurements were normalized to AAV9. **f** A 100K variant Fit4Function library was injected intravenously into a cynomolgus macaque and the vector genome distribution was assessed four hours later. Variants predicted to meet all six trait conditions were highly enriched in the cynomolgus macaque liver (biodistribution). The density plot shows the distribution of variants normalized to the sum of counts for each indicated set of variants. **g** The fraction of the indicated MultiFunction variants enriched in the cynomolgus macaque liver (defined as at least two-fold log₂ enrichment greater than that of AAV9) are shown for each combination of predicted traits. Binding and transduction are indicated by 'b' and 't', respectively. **h** The rhesus macaque liver transduction efficiency, measured by transcript levels 4-weeks post-administration, for the MultiFunction variants are shown (*n* = 2 rhesus macaques). Each variant was represented by two codon replicates while AAV9 was represented by three codon replicates. Source data are provided in a Source Data file.

Our finding that six of seven individually tested MultiFunction liver-enriched variants exhibited reduced targeting of the brain, spinal cord, and kidney compared to AAV9 (Fig. 4d) suggested that the models did not simply learn to predict variants with generally improved uptake into cells. To gain insight into the features learned by the models, we investigated how each amino acid residue at each position in the 7-mer, different physicochemical properties at each position, as well as epistatic interactions contribute to explaining the signals learned by each of the models. Across the six models, we observed that certain charged residues had a large overall impact on

the predicted scores regardless of which position they occupied within the 7-mer; the presence of Cysteine or Tryptophan at any position led to reduced production fitness predicted scores, and the presence of Arginine, Lysine, or Cysteine tended to lead to predictions of increased liver cell targeting (Supplementary Fig. 9a). Positive and negative charges contributed to increases and decreases in predicted scores, respectively (Supplementary Fig. 9b). However, physicochemical properties alone could not fully account for the signal learned by the models (Supplementary Fig. 9b). The models learned largely from both first-degree (single residue) and second-degree (paired residues) interactions but those interactions could not fully explain the original model's predictions (1st and 2nd degree interactions explain 0.37–0.78 and 0.78–0.94 of the model predictions, respectively; Supplementary Fig. 9c). This indicates that other complex signals or higher order epistasis contribute to the predictions of the original models.

### Fit4Function translates across species to macaques

We administered a 100K-member Fit4Function library intravenously to an adult cynomolgus macaque and assessed biodistribution. Liver-targeted MultiFunction capsids, predicted with the six prior models that were trained only on human cell and mouse data and production fitness, were highly enriched in terms of macaque liver biodistribution (Fig. 4f). The combination of multiple functional predictors was more effective at identifying variants with increased biodistribution to the macaque liver than any predictor used in isolation (Fig. 4g). The five liver models exhibited redundancy, which is unsurprising given that they are readouts of related functions (Fig. 4g). Notably, the in vitro human hepatocyte transduction models translated better to cynomolgus macaque liver biodistribution compared to the in vivo mouse liver biodistribution model, which was neither necessary nor sufficient to demonstrate transferability to cynomolgus macaque liver biodistribution; the hit rate did not decrease when the mouse liver model was excluded from the combination of models (Fig. 4g). The hit rate decreased only modestly when both human hepatocyte transduction models were excluded, demonstrating the utility of using models in combination (Fig. 4g). Separately, we performed a transduction study in rhesus macaques and found that six of the seven liver MultiFunction capsids individually validated in human cells and in mice in vivo (Fig. 4d, Supplementary Fig. 7) were more efficient than AAV9 at transducing the rhesus macaque liver when administered as a library (Fig. 4h; $n = 2$ rhesus macaques, BI152 was inadvertently excluded due to a library assembly error).

## Discussion

The Fit4Function pipeline presents a significant conceptual and technological advance over prior AAV engineering studies, including those that leverage ML. Conventional in vivo selections use sequential rounds to narrow the focus of sequence exploration to a handful of top candidates, which may not have other traits required for translation to preclinical models and clinical trials. Simultaneously engineering multiple traits into AAV capsids or other proteins of interest is an important but challenging goal. To date, most protein engineering efforts, including those leveraging ML, have focused on optimizing a single function, e.g., generating more efficiently produced and diversified AAV capsid libraries but stopping short of multi-trait prediction[13,17,18]. A few groups have gone beyond single trait engineering by combining multiple previously validated functional structures into a single protein, e.g., by recombining structurally independent segments from different channelrhodopsins possessing known functions, localizations, and photocurrent properties of interest[19], or by applying protein design tools to filter out variants that do not meet additional characteristics such as solubility and immunogenicity[20]. However, as these strategies rely on the recombination of multiple existing functional structures into a single protein or the use of third-party protein design tools, they cannot be broadly

generalized to engineer multiple de novo functions. A key obstacle to combining multiple ML models that predict different traits is the aggregated error that increases with each added model. The Fit4-Function approach directly tackles this problem by leveraging a moderately sized, all viable, low-bias (ML-designed) library to generate highly reproducible data for multi-trait learning with a low false positive rate. This allows the models to be applied in different combinations with a low risk of aggregating significant error. MultiFunction libraries can thus be generated to more efficiently explore the vast sequence space for multi-trait capsids.

The Fit4Function approach can help to reduce the need for extensive screening in animals. Firstly, the distinct features of Fit4-Function libraries enable the quantitative assessment of capsid biodistribution and candidate selection from just a single round of screening. It is only necessary to screen a Fit4Function library once for a given function to then predict the functionality of sequences that were not contained in the original library. In contrast, it typically requires two or more rounds of in vivo screening to reliably identify top candidates from conventional selections, and the data from these screens cannot be used to accurately predict the traits of variants not tested in that screen. This means that the Fit4Function approach can be used to design libraries full of diverse and promising candidates for more efficient screening in animals or in vitro assays. Secondly, unlike existing screening strategies, our approach can systematically determine the functional assays or combinations thereof that drive cross-species transferability. As the Fit4Function approach is applied to more functions of interest (e.g., crossing the blood-brain barrier), it will become apparent whether it is worthwhile to continue screening in mice or other animals for those functions. This can inform the choice of cell or animal models used to perform screens and to develop vectors that are more likely to translate preclinically and clinically.

As with other ML-guided approaches, Fit4Function can be more challenging to implement with assays that produce low-quality data due to lower detection sensitivities. For example, data reproducibility and subsequent model performance can be bottlenecked by in vivo transduction assays in some organs due to the inherent tropism of the parental capsid, inter-animal variability, and technical challenges related to tissue sampling in small animals or in organs where tissues are limited. One approach to improve data quality with low sensitivity assays may be to use smaller Fit4Function libraries because reducing library diversity increases the sampling of each individual variant and therefore the quality of the screening data. A second limitation that affects any multi-objective engineering effort, the Fit4Function approach included, is that variants that are maximally optimized for multiple objectives may not exist, especially in cases where performance on functions are negatively correlated (see Supplementary Fig. 4 for examples).

With continued application across experiments and laboratories, the Fit4Function approach should enable the assembly of a vast ML atlas that can accurately predict the performance of AAV capsid variants across dozens of traits and inform the design of screening pipelines. In addition, the Fit4Function approach should translate to engineering other proteins that are amenable to quantitative, high-throughput screening of libraries that are diversified at a defined set of residues.

## Methods

### Ethical Statement

All mouse procedures were performed as approved by the Broad Institute Institutional Animal Care and Use Committee (IACUC), approval number 0213-06-18-1. For the cynomolgus macaque experiment ($n = 1$), the study plan involving the care and use of animals was reviewed and approved by the Charles River CR-LAV Institutional Animal Care and Use Committee (IACUC). The rhesus macaque study ($n = 2$) was conducted at the NIH Nonhuman Primate Testing Center

for Evaluation of Somatic Cell Genome Editing Tools at the University of California, Davis (UC Davis). All procedures conformed to the requirements of the Animal Welfare Act, and protocols were approved prior to implementation by the UC Davis IACUC.

## Animals

All mouse procedures were performed as approved by the Broad Institute Institutional Animal Care and Use Committee (IACUC), approval number 0213-06-18-1. Unless otherwise stated, all mice were female C57BL/6J (000664) mice from the Jackson Laboratory (JAX). During individual variant characterization, recombinant AAV vectors were administered intravenously via the retro-orbital sinus in young adult female (7–8-week-old) C57BL/6J animals ($n = 5$ mice per vector group). Mice were randomly assigned to groups based on pre-determined sample sizes. No mice were excluded from the analyses. For all assays, mice were euthanized with EUTHASOL™ (Virbac) and transcardially perfused with phosphate buffer saline, pH 7.4, at room temperature (RT). Experimenters were not blinded to the sample groups.

For the cynomolgus macaque experiment ($n = 1$), the study plan involving the care and use of animals was reviewed and approved by the Charles River CR-LAV Institutional Animal Care and Use Committee (IACUC). During the study, the care and use of animals were conducted by CR-LAV with guidance from the USA National Research Council and the Canadian Council on Animal Care (CCAC). The Test Facility is accredited by the CCAC and AAALAC. Per the CCAC guidelines, this study was considered as a category of invasiveness C.

The rhesus macaque study ($n = 2$) was conducted at the NIH Nonhuman Primate Testing Center for Evaluation of Somatic Cell Genome Editing Tools at the University of California, Davis (UC Davis). All procedures conformed to the requirements of the Animal Welfare Act, and protocols were approved prior to implementation by the UC Davis IACUC.

## Modeling and assessment library design

The modeling and assessment libraries were designed to contain 150K nucleotide sequences each. The libraries were composed of 64.5K unique and 10K shared amino acid sequences generated by uniformly sampling all 20 amino acids at each position. The combined 74.5K variants were duplicated via codon replication. 1K nucleotide sequences containing stop codons were included to detect any potential problems with cross-packaging.

## Capsid library synthesis

To produce synthetic library inserts, lyophilized DNA oligonucleotide libraries (Agilent G7223A) or NNK hand-mixed primers (IDT) were spun down at 8000 RCF for 1 min, resuspended in 10 μL UltraPure DNase/RNase-Free Distilled Water (Thermo Fisher Scientific, 10977015), and incubated at 37 °C for 20 min. For pooled synthetic oligonucleotide libraries, the following primer format was used: 5′-GTATTCCTTGG TTTTGAACCCAACCGGTCTGCGCCTGTGC-(NNN)$_7$-TTGGGCACTCTG GTGGTTTGTGGCCAC-3′. To produce NNK inserts, the AAV9_K449R_-Forward (5′-CGGACTCAGACTATCAGCTCCC-3′) and AAV9_K449R_ NNK_Reverse (5′-GTATTCCTTGGTTTTGAACCCAACCGGTCTGCGCCT GTGC-(MNN)$_7$-TTGGGCACTCTGGTGGTTTGTG-3′) primers were used.

To amplify the oligonucleotide libraries and incorporate them into an AAV9 (K449R) template, 2 μL of the resuspended pooled oligonucleotide library or NNK-based library was used as an initial reverse primer along with 0.5 μM AAV9_K449R_Forward primer in a 25 μL PCR amplification reaction using Q5 Hot Start High-Fidelity 2× Master Mix (NEB, M0494S). 50 ng of a plasmid containing only AAV9 (K449R) VP1 amino acids 347–586 was used as a PCR template. PCR was performed following the manufacturer's protocol with an annealing temperature of 65 °C for 20 s and an extension time of 90 s. After six PCR cycles, 0.5 μM AAV9_K449R_Reverse (5′-GTATTCCTTGGTTTTGAACCCAACCG-3′) was

spiked into the reaction as a reverse primer to further amplify sequences containing the oligonucleotide library for an additional 25 cycles. To remove the PCR template, 1 μL of DpnI (NEB, R0176S) was added to the PCR reaction and incubated at 37 °C for 1 h. Afterward, the PCR products were cleaned using AMPure XP beads (Beckman, A63881) following the manufacturer's protocol.

The PCR insert was assembled into 1600 ng of a linearized mRNA selection vector (AAV9-CMV-Express) with NEBuilder HiFi DNA Assembly Master Mix (NEB, E2621L) at a 3:1 insert:vector Molar ratio in an 80 μL reaction volume, incubated at 50 °C for 1 h, and then at 72 °C for 5 min. Afterward, 4 μL of Quick CIP (NEB, M0508S) was spiked into the reaction and incubated at 37 °C for 30 min to dephosphorylate unincorporated dNTPs that may inhibit downstream processes. Finally, 4 μL of T5 Exonuclease (NEB M0663S) was added to the reaction and incubated at 37 °C for 30 min to remove unassembled products. The final assembled products were cleaned using AMPure XP beads (Beckman, A63881) following the manufacturer's protocol and their concentrations were quantified with a Qubit dsDNA HS Assay Kit (Thermo Fisher Scientific, Q32851) and a Qubit fluorometer.

## mRNA selection vector

The mRNA selection vector (AAV9-CMV-Express) was designed to enrich for functional AAV capsid sequences by recovering capsid mRNA from transduced cells. AAV9-CMV-Express uses a ubiquitous CMV enhancer and AAV5 p41 gene regulatory elements to drive AAV Cap expression. The AAV-Express plasmid was constructed by cloning the following elements into an AAV genome plasmid in the following order: a cytomegalovirus (CMV) enhancer-promoter, a synthetic intron, and the AAV5 P41 promoter along with the 3′ end of the AAV2 Rep gene, which includes the splice donor sequences for the capsid RNA. The capsid gene splice donor sequence in AAV2 Rep was modified from a non-consensus donor sequence CAGGTACCA to a consensus donor sequence CAGGTAAGT. The AAV9 capsid gene sequence was synthesized with nucleotide changes at S448 (TCA to TCT, silent mutation), K449R (AAG to AGA), and G594 (GGC to GGT, silent mutation) to introduce restriction enzyme recognition sites for oligonucleotide library fragment cloning. The AAV2 polyadenylation sequence was replaced with a simian virus 40 (SV40) late polyadenylation signal to terminate the capsid RNA transcript.

## AAV production

For library production, HEK293T/17 cells (ATCC, CRL-11268) were seeded at 22 million cells per 15 cm plate the day before transfection and grown in DMEM with GlutaMAX (Gibco, 10569010) supplemented with 5% FBS and 1× non-essential amino acid solution (NEAA) (Gibco, 11140050). The next day, each plate was triple transfected with 39.93 μg of total plasmid DNA encompassing pHelper, RepStop encoding the AAV2 Rep genes, and pUC19 at a ratio of 2:1:1, respectively, alongside 10 ng of assembled library DNA. The media was exchanged for fresh DMEM with 5% FBS and 1× NEAA at 20 h post-transfection. At 60 h post-transfection, the media and cell lysates were harvested and purified following the published protocol[21].

Individual recombinant AAVs were produced in suspension HEK293T cells, using F17 media (Thermo Fisher Scientific). Cell suspensions were incubated at 37 °C, 8% $CO_2$, 125 RPM. At 24 h before transfection, cells were seeded into 200 mL of media at -1 million cells/mL. The day after, the cells (-2 million cells/mL) were transfected with pHelper, RepCap, and transgene plasmids (2:1:1 ratio, 2 μg DNA per million cells) using Transport 5 transfection reagent (Polysciences) with a 2:1 PEI:DNA ratio. At three days post-transfection, cells were pelleted at 2000 RPM for 10 min into Nalgene conical bottles. The supernatant was discarded, and cell pellets were stored at −20 °C until purification. Each pellet, corresponding to 200 mL of cell culture, was resuspended in 7 mL of 500 mM NaCl, 40 mM Tris-base, and 10 mM $MgCl_2$, with Salt Active Nuclease (ArcticZymes, #70920-202) at 100 U/

mL. Afterward, the lysate was clarified at 2000 RCF for 10 min and loaded onto a density step gradient containing OptiPrep (Cosmo Bio, AXS-1114542) at 60%, 40%, 25%, and 15% at a volume of 5, 5, 6, and 6 mL respectively in OptiSeal tubes (Beckman, 361625). The step gradients were spun in a Beckman Type 70ti rotor (Beckman, 337922) in a Sorvall WX+ ultracentrifuge (Thermo Fisher Scientific, 75000090) at 340,252 × *g* for 1 h at 18 °C. Afterward, ~4.5 mL of the 40–60% interface was extracted using a 16-gauge needle, filtered through a 0.22 μm PES filter, buffer exchanged with 100K MWCO protein concentrators (Thermo Fisher Scientific, 88532) into PBS containing 0.001% Pluronic F-68, and concentrated down to a volume of 500 μL. The concentrated AAV was filtered through a 0.22 μm PES filter and stored at 4 °C or −80 °C.

## AAV titering

To determine AAV titers, 5 μL of each purified AAV library were incubated with 100 μL of an endonuclease cocktail consisting of 11,000 U/mL Turbonuclease (Sigma T4330-50KU) with 1× DNase I reaction buffer (NEB B0303S) in UltraPure DNase/RNase-Free distilled water at 37 °C for 1 h. Next, the endonuclease solution was inactivated by adding 5 μL of 0.5 M EDTA, pH 8.0 (Thermo Fisher Scientific, 15575020) and incubated at room temperature for 5 min and then at 70 °C for 10 min. To release the encapsidated AAV genomes, 120 μL of a Proteinase K cocktail consisting of 1 M NaCl, 1% N-lauroylsarcosine, 100 μg/mL Proteinase K (Qiagen, 19131) in UltraPure DNase/RNase-Free distilled water was added to the mixture and incubated at 56 °C for 2–16 h. The Proteinase K-treated samples were then heat-inactivated at 95 °C for 10 min. The released AAV genomes were serial diluted between 460–460,000× in dilution buffer consisting of 1× PCR Buffer (Thermo Fisher Scientific, N8080129), 2 μg/mL sheared salmon sperm DNA (Thermo Fisher Scientific, AM9680), and 0.05% Pluronic F-68 (Thermo Fisher Scientific, 24040032) in UltraPure Water (Thermo Fisher Scientific). Then, 2 μL of the diluted samples were used as input in a ddPCR supermix (Bio-Rad, 1863023). Primers and probes, targeting the ITR and CAG promoter region, were used for titration, at a final concentration of 900 nM and 250 nM, respectively (ITR2_Forward: 5′-GGAACCCCTAGTGATGGAGTT-3′; ITR2_Reverse: 5′-CG GCCTCAGTGAGCGA-3′; ITR2_Probe: 5′-CACTCCCTCTCTGCGCGCTCG-3′ [FAM/Iowa Black FQ Zen]; CAG_Forward: 5′-TGTTCCCATAGTAACG CCAATAG-3′; CAG_Reverse: GTACTTGGCATATGATACACTTGATG-3′; CAG_Probe: 5′-TTACGGTAAACTGCCCACTTGGCA-3′ [FAM/Iowa Black FQ Zen]). Droplets were generated using a QX100 Droplet Generator following the manufacturer's protocol. The droplets were transferred to a thermocycler and cycled according to the manufacturer's protocol with an annealing/extension of 58 °C for 1 min. Finally, droplets were read on a QX100 Droplet Digital System to determine titers.

## Assessing production fitness

To recover only encapsidated AAV genomes for downstream analysis, $10^{11}$ viral genomes were extracted using the endonuclease and Proteinase K steps outlined above (AAV Titering). After Proteinase K treatment, samples were column purified using a DNA Clean and Concentrator Kit (Zymo Research, D4033) and eluted in 25 μL elution buffer for NGS preparation.

## NGS sample preparation

To prepare AAV libraries for sequencing, qPCR was performed on the extracted AAV genomes or cDNA to determine the cycle thresholds for each sample type to prevent overamplification. PCR amplification using equal primer pairs (1–8) (described in ref. 22) was used to attach partial Illumina Read 1 and Read 2 sequences using Q5 Hot Start High-Fidelity 2× Master Mix with an annealing temperature of 65 °C for 20 s and an extension time of 60 s. Round one PCR products were purified using AMPure XP beads following the manufacturer's protocol and eluted in 25 μL UltraPure Water (Thermo Fisher Scientific). Then, 2 μL was used as input in a second round of PCR to attach on Illumina

adapters and dual index primers (NEB, E7600S) for five PCR cycles using Q5 Hot Start-High-Fidelity 2× Master Mix with an annealing temperature of 65 °C for 20 s and an extension time of 60 s. The round two PCR products were purified using AMPure XP beads following the manufacturer's protocol and eluted in 25 μL UltraPure DNase/RNase-Free distilled water (Thermo Fisher Scientific).

To quantify the amount of PCR products for NGS, an Agilent High Sensitivity DNA Kit (Agilent, 5067-4626) was used with an Agilent 2100 Bioanalyzer. PCR products were pooled and diluted to 2–4 nM in 10 mM Tris-HCl, pH 8.5 and sequenced on an Illumina NextSeq 550 following the manufacturer's instructions using a NextSeq 500/550 Mid or High Output Kit (Illumina, 20024904 or 20024907), or on an Illumina NextSeq 1000 following the manufacturer's instructions using NextSeq P2 v3 kits (Illumina, 20046812). Reads were allocated as follows: I1: 8, I2: 8, R1: 150, R2: 0.

## NGS data processing

Sequencing data was de-multiplexed with *bcl2fastq* (version v2.20.0.422) using the default parameters. The Read 1 sequence (excluding Illumina barcodes) was aligned to a short reference sequence of AAV9:

CCAACGAAGAAGAAATTAAAACTACTAACCCGGTAGCAACGGAG TCCTATGGACAAGTGGCCACAAACCACCAGAGTGCCCAAN̲N̲N̲N̲N̲N̲N̲N̲ N̲N̲N̲N̲N̲N̲N̲N̲N̲N̲N̲N̲N̲N̲GCACAGGCGCAGACCGGTTGGGTTCAAAACC AAGGAATACTTCCG

Alignment was performed with *bowtie2* (version 2.4.1)[23] with the following parameters:

--end-to-end --very-sensitive --np 0 --n-ceil L,21,0.5 --xeq -N 1 --reorder --score-min L,-0.6,-0.6 -5 8 -3 8

Resulting sam files from *bowtie2* were sorted by read and compressed to bam files with *samtools* (version 1.11-2-g26d7c73, *htslib* version 1.11-9-g2264113)[24,25].

*Python* (version 3.8.3) scripts and *pysam* (version 0.15.4) were used to extract the 21-nucleotide insertion from each amplicon read. Each read was assigned to one of the following bins: Failed, Invalid, or Valid. Failed reads were defined as reads that did not align to the reference sequence, or that had an indel in the insertion region (i.e., 20 bases instead of 21 bases). Invalid reads were defined as reads whose 21 bases were successfully extracted but matched any of the following conditions: (1) Any one base of the 21 bases had a quality score (AKA Phred score, QScore) below 20, i.e., error probability >1/100, (2) any one base was undetermined, i.e., "N", (3) the 21 base sequence was not from the synthetic library (this condition does not apply to NNK library variants). Valid reads were defined as reads that did not fit into either the Failed or Invalid bins. The Failed and Invalid reads were collected and analyzed for quality control purposes, and all subsequent analyses were performed on the Valid reads.

Count data for valid reads was aggregated per sequence, per sample, and was stored in a pivot table format, with nucleotide sequences on the rows, and samples (Illumina barcodes) on the columns. Sequences not detected in samples were assigned a count of 0.

## Data normalization

Count data was reads-per-million (RPM) normalized to the sequencing depth of each sample (Illumina barcode) with:

$$r_{i,j} = \frac{k_{i,j}}{\sum_{l=1}^{n} k_{l,j}} \times 1,000,000 \qquad (1)$$

Where *r* is the RPM-normalized count, *k* is the raw count, *i = 1 … n* sequences, and *j = 1 … m* samples.

As each biological sample was run in triplicate, we aggregated data for each sample by taking the mean of the RPMs across *p*

replicates of sample $s$:

$$\mu_{i,s} = \frac{\sum_{l=1}^{p} r_{i,l}}{p} \qquad (2)$$

We estimated normalized variance across replicates by taking the coefficient of variation (CV):

$$CV_{i,s} = \frac{\mu_{i,s}}{\sigma_{i,s}} \qquad (3)$$

where $\sigma_{i,s}$ is the standard deviation for variant $i$ in sample $s$ over $p$ replicates.

Log$_2$ enrichment for each sequence was defined as:

$$e_{i,s} = \log_2\left(\frac{\mu_{i,s}}{\mu_{corrected,i,t}}\right) \qquad (4)$$

where $e$ is the log$_2$ enrichment, $\mu$ is the mean of the replicate RPMs, and $t$ is the normalization sample. For production fitness, the sample $s$ is the variant abundance after AAV production, and the normalization sample $t$ is the variant abundance in the plasmid pool. For functional screens, the sample $s$ is the variant abundance of the screen, and the normalization factor $t$ is the variant abundance after AAV production.

To avoid dividing by 0 in $e$ (for NNK library processing), $\mu_{corrected}$ is defined as:

$$\mu_{corrected,i,t} = \begin{cases} \mu_{i,t}, & \mu_{i,t} > 0 \\ \frac{1}{\sum_{l=1}^{n} k_{l,t}}, & \mu_{i,t} = 0 \end{cases} \qquad (5)$$

Counts of 0 across all three replicates for the normalization sample were adjusted to a pseudocount of 1 across all three replicates.

## Production fitness training and assessment

We designed a robust ML framework for the production fitness and Fit4Function functional mappings. A long short-term memory (LSTM) regression model with two hidden layers of 140 and 20 nodes was implemented in *Keras*[26]. RNNs, and LSTMs in particular, have been successfully applied for learning functions from biological sequence data as they are designed to capture local and distant relationships across different parts of the input sequences[13,27]. Model parameters and hyperparameters were subject to fine-tuning processes but no significant performance was gained across all different functional fitness models implemented in this study. Therefore, we kept a simple model architecture across all modeling throughout this study. The input layer was 7-mer amino acid sequences one-hot encoded into a $20 \times 7$ matrix. The target/output is the relative production or functional fitness score. The loss was optimized by mean-squared error with the Adam optimizer running on a learning rate of 0.001[28]. The batch size was set to 500 observations. To avoid overfitting, model training was controlled by a custom early stopping procedure where the training process was terminated if the ratio of training error to validation error dropped below 0.90.

For production fitness learning, the training size was optimized by training the framework on increments of 1K variants. Variants that were not detected ($n = 5693$) after AAV production were filtered out from training. Model validation performance was reported at each training size, and a size of 24K variants was arbitrarily selected for final model training given that the model performance reached a plateau after a training size of ~5K. The modeling library core variants ($N$ ~ 60K after removing the non-detected sequences) were then randomly divided into training (24K), validation (5K), and testing subsets (30.6K), all from the modeling library. The model was trained on the training set, validated during the training process on the validation set, and tested on the testing set. The model was further tested on the unique variants from the assessment library to assess its generalization across libraries.

## Fit4Function library sampling

The Fit4Function libraries are intended to be sampled from the production-fit space. For the Fit4Function library utilized in this study, we first uniformly sampled a set of 7-mer amino acid sequences 100× the required library size (240K Fit4Function variants * 100 = 24M variants), by equally sampling each amino acid at each of the seven positions. Duplicates were removed and the remaining sequences were scored using the production fitness model. Then, the 240K Fit4Function library variants were probabilistically sampled from the parametrized production-fit distribution. In addition to the 240K production-fit variants, we added 1K stop-codon-containing variants and 3K variants from the 10K variants shared between the modeling and assessment libraries as a control set.

## Fit4Function library validation

Fitness enrichment scores are relative across library variants due to normalization calculations; calibration is needed to make the fitness scores of libraries of different compositions comparable for assessment or integration purposes. Therefore, we used the 3K control set in the Fit4Function library to fit an ordinary linear regression model of the measured production fitness scores between the Fit4Function library and the modeling library. These regression parameters were applied to the production fitness measured scores of the 240K Fit4Function variants to obtain calibrated production fitness scores. After synthesizing the Fit4Function library, we compared, by means of correlation, the predicted fitness scores to the calibrated measured fitness.

## AAV mouse in vivo biodistribution assays

Purified AAV libraries were injected intravenously (retro-orbital sinus) into four C57BL/6J adult female mice at a dose of $1 \times 10^{12}$ vg/mouse (6–8 weeks of age). Two hours post-injection, serum was collected and tissues were harvested using disposable 3 mm biopsy punches (Integra, 33-32-P/25) with a new biopsy punch used per organ per replicate. Harvested tissues were immediately frozen on dry ice. AAV genomes were recovered using a DNeasy kit (Qiagen, 69504) following the manufacturer's protocol, and samples were eluted in 200 μL elution buffer for NGS preparation.

## AAV cynomolgus macaque in vivo biodistribution assays

The library administered had 100K unique amino acid variants following the Fit4Function criteria (uniformly sampled from the production-fit sequence space) in addition to a calibration set (3K), control variants, and AAV9. Each variant in the Fit4Function distribution was represented by either two or six codon replicates; AAV9 was represented by two codon replicates. The purified AAV library was injected intravenously at a dose of $4.6 \times 10^{12}$ vg/kg into a 3.4 kg adult female cynomolgus macaque that was pre-screened for neutralizing antibodies (NAb) against AAV9 (CRL). Four hours after systemic delivery, the animal was sedated with an intramuscular injection of a combination of ketamine hydrochloride and acepromazine, and anesthetized by a mixture of isoflurane and oxygen. Next, the animal was perfused with 0.9% sodium chloride, and tissues were harvested and snap-frozen on dry ice. DNA was extracted using a DNeasy kit in a Qiagen QIAcube Connect. Samples were then processed as detailed in the "NGS sample preparation" section.

## AAV rhesus macaque in vivo transduction assays

Approximately 3-month-old rhesus macaques (~1 kg; one male, one female) were screened and assigned to the project after confirming seronegative status for AAV9 antibodies using standardized Testing Center assays. Sedation with Telazol (IM) was performed prior to

intravenous administration of a purified AAV library ($1 \times 10^{13}$ vg/kg) with blood samples collected (~4 mL; hematology, clinical chemistry, serum, plasma; pre-administration and weekly post-administration). Animals were monitored closely during the study period and until the endpoint (four weeks post-administration) and remained robust and healthy with no evidence of adverse findings. Four weeks after systemic delivery, tissues were collected and snap-frozen over liquid nitrogen then placed on dry ice immediately prior to storage at $\leq -80\,°C$. RNA and DNA were extracted using TRIzol (Invitrogen, 15596026) following the manufacturer's instructions. Total RNA was then processed through an RNeasy kit (Qiagen, 74106) followed by on-column DNA digestion. RNA was converted to cDNA using Maxima H Minus Reverse Transcriptase (Thermo Fisher Scientific, EP0751) according to the manufacturer's instructions. Samples were then processed as detailed in the "NGS sample preparation" section.

### Rhesus macaque serum screening for anti-AAV9 neutralizing antibodies

Neutralization assays were performed at 500 or 1000 vector genomes (vg)/cell in Perkin–Elmer white 96-well plates. Four-fold serial dilutions (1:4 to 1:16,384) of macaque serum samples were prepared in 96-well plates using DMEM supplemented with 5% fetal bovine serum (FBS). Then, 40 μL of each dilution was transferred to a separate 96-well plate, mixed with an equal volume of AAV9:CAG-GFP-P2A-Luciferase-WPRE-SV40 vector ($4–8 \times 10^7$ vg per 40 μL, diluted in DMEM-5% FBS), and incubated for 1 h at 37 °C. Following the incubation, AAV-serum samples were transferred into a new 96-well plate (20 μL triplicates) and a total of 80 μL of DMEM-5% fetal bovine serum, containing 20,000 HEK293T cells, was added to each well (final volume of 100 μL). 96-well plates were incubated for 48 h at 37 °C, 5% $CO_2$. Luminescence levels were read using a Perkin–Elmer Victor Luminescence Plate Reader using the britelite plus Reporter Gene Assay System (Perkin–Elmer, #6066761). Data was analyzed using the *neutcurve* Python package developed by the Bloom laboratory. The NAb titer was measured as the concentration that resulted in a 50% reduction in luciferase activity relative to the no-serum control. Animals used in the transduction study had NAb titers of <1:12 in this set of antibody screens.

### In vitro binding and transduction

HEK293T/17 (ATCC® CRL-11268™), HEPG2 (ATCC® HB-8065™), THLE-2 (ATCC® CRL-2706™), hCMEC/D3 (Millipore, SCC066), and human and mouse BMVECs (Cell Biologics, H-6023 and C57-H6023) were used in this study. Among these, HEK293T/17 is the only cell line known to be potentially cross-contaminated with HeLa cells[29]. We used morphology checks under light microscopy with a Leica DM IL LED Inverted Laboratory Microscope to rule out cross-contamination with HeLa cells. Cells were grown in 100 mm dishes and exposed to the Fit4-Function or NNK 7-mer library ($1 \times 10^4$ vg/cell for HEK293T/17, $3 \times 10^4$ vg/cell for hCMEC/D3, $6 \times 10^4$ vg/cell for primary human and mouse BMVECs, and $5 \times 10^3$ vg/cell for HEPG2 and THLE-2) diluted in 10 mL of growth media at 4 °C with gentle rocking for 2 h. Cells were then washed 3× with DPBS, and total DNA was extracted with the DNeasy kit (Qiagen) according to the manufacturer's instructions. Half of the recovered DNA was used in PCR amplification for viral genome sequence recovery.

Transduction assays were performed as described above with the following exceptions: The cells were cultured in growth media containing AAV for 60 h and total RNA was then extracted with the RNeasy kit (Qiagen). From the total RNA, 5 μg was converted to cDNA using the Maxima H Minus Reverse Transcriptase according to the manufacturer's instructions.

### Sequence-to-function mapping

Functional scores were quantified as the $\log_2$ of the fold-change enrichment of the variant reads-per-million (RPM) after the screen relative to its RPM in the AAV library, i.e., $\log_2$ (Assay RPM/Virus RPM). Fit4Function models utilized the same design of the ML framework utilized for production fitness mapping (two-layer LSTM, custom early stopping, batch size of 500 variants, MSE error, and Adam optimizer). Out of the 240K variants in the Fit4Function library, 90K were allocated for training and testing the ML function models (model construction), and 150K variants were held out for validation of the MultiFunction approach. The training size for each model was optimized independently. As with the production fitness model, the liver-related functional fitness models were assessed by correlation between the predicted and measured functional scores.

### MultiFunction library design

Using the previously generated fitness models of the production fitness and the five liver-related functional fitness models described in the main text, we conducted an in silico screen of 10M randomly sampled 7-mer sequences to identify variants that are highly fit for all six traits. The threshold of high fitness for each function was arbitrarily set to the 50th percentile of each functional fitness distribution from the Fit4Function screening data. The percentiles were calculated on the detected variants of each functional assay from the 90K model construction data set. To reduce false positive predictions, we arbitrarily increased the threshold for predicted hits by 5% of the fitness dynamic range for each function. These thresholds were used to filter the 10M variants that were run through the six functional prediction models for variants predicted to pass the six modified thresholds. Of these, we sampled 30K variants to be included in the MultiFunction library. The 30K variants were each represented by two codon replicates. The MultiFunction library also included (1) a positive control set (3K) that was drawn from the subset of the 150K Fit4Function validation set that met the six conditions on the actual measurements, (2) a set of 10K variants randomly sampled from the Fit4Function 240K core variants as background controls representing the production-fit space, (3) a set of 3K calibration variants present in both the Fit4Function library and the modeling library to be used as background controls representing the entire (unbiased) sequence space, and (4) 1K stop-codon-containing sequences.

### MultiFunction library validation

The MultiFunction library was synthesized, AAV was produced, and the five liver-related functions were screened for as was done with the Fit4Function library. We quantified the success rate of the Multi-Function library in terms of hit rate, i.e., out of the 30K variants predicted to meet the six criteria, what percentage satisfied the six criteria when the MultiFunction library was screened on those functions (predicted positive versus measured positive). To determine whether a variant meets specific functional criteria, we compared the distribution of that function for the MultiFunction variants against the positive control set. For a variant to be considered a hit for a given function, its measured enrichment score had to be greater than the mean − 2 SD (standard deviations) of the enrichment scores of the positive control set measured in the same experiment. This thresholding was applied to avoid the overestimation of the hit rate due to outliers in the positive control set. A variant is considered a hit in calculating the Multi-Function hit rate only if it is a hit for all six functions; a variant that meets only five or fewer conditions is not considered a hit.

As the sample sizes for the Fit4Function and the Uniform control sets in the MultiFunction library are small (3K and 10K, respectively), we estimated the multi-trait hit rate from the 240K Fit4Function library to provide a more confident assessment of the percentage of variants in the production-fit space and the uniform space that meet all six criteria. We calculated the hit rate of the Fit4Function space as the percentage of the production-fit variants that are measured to pass the thresholds for all six functions simultaneously (17.14K out of the 240K Fit4Function variants). The thresholds used

are the same as those used to define the multi-trait variant set from which the positive control set was drawn. As the uniform libraries in the study (the Modeling and Assessment libraries) were not used in screens for the five liver-related functions, we could not use them to assess the Uniform hit rate. Instead, we estimated the hit rate for the uniform space as the hit rate in the production-fit space scaled by the percentage of the uniform space occupied by the production-fit space. In other words, Uniform hit rate = Fit4Function hit rate × production-fit ratio = 7.0% × 37.3% = 2.6%, given that no non-fit variants meet the criteria for production fitness. For comparison, when the Uniform multi-trait hit rate was calculated from the small Uniform control sets in the Fit4Function and the MultiFunction libraries, we obtained hit rates of 0.0% and 4.6%, respectively. When the Fit4Function hit rate was calculated from the Fit4Function control set in the MultiFunction library, we obtained a hit rate of 15.4%. We believe that the hit rates calculated with the larger Fit4Function library are more accurate.

### Individual capsid characterization

Individual capsids were cloned into iCAP-AAV9 (K449R) backbone (GenScript), produced as described above, and administered to C57BL/6J (The Jackson Laboratory, 000664) mice at a dose of $1 \times 10^{10}$ vg/mouse ($n = 5$/group). Three weeks later, three separate lobes of the liver were collected for RNA extraction and a single lobe per mouse was fixed in 4% PFA.

For microscopy, fixed liver tissues were sectioned at 100 μm using a Leica VT1200 vibratome. Sections were mounted with ProLong™ Gold Antifade Mountant with DAPI (Thermo Fisher Scientific, P36931). Liver images were collected using the optical sectioning module on a Keyence BZ-X800 with a Plan Apochromat 20× objective (Keyence, BZ-PA20). Three images were taken for each animal ($n = 5$/group) and compared to a no-injection control ($n = 3$ animals). In CellProfiler, nuclei were segmented and DAPI$^+$ nuclei were identified using a threshold on DAPI intensity determined from the no-injection control. Each DAPI$^+$ nuclei was then quantified with the median pixel intensity in the GFP channel.

For assessment of liver transduction by quantitative RT-PCR, total RNA was recovered using TRIzol (Invitrogen, 15596026) following the manufacturer's instructions. Total RNA was then processed (RNeasy kit, Qiagen, 74106) followed by on-column DNA digestion. RNA was converted to cDNA using Maxima H Minus Reverse Transcriptase (Thermo Fisher Scientific, EP0751) according to manufacturer instructions. Afterward, qPCR was used to detect AAV-encoded RNA transcripts with the following primer pair (5′-GCACAAGCTGGAGTA-CAACTA-3′) and (5′-TGTTGTGGCGGATCTTGAA-3′) and the following primer pair for GAPDH (5′-ACCACAGTCCATGCCATCAC-3′) and (5′-TCCACCACCCTGTTGCTGTA-3′).

THLE-2 and HEPG2 cells were seeded in a 96-well plate the day before adding the AAVs at 5000 vg/cell. For binding assays, AAVs were diluted in media and incubated with cells at 4 °C with gentle shaking for 1 h. After incubation, cells were washed 3× with PBS to remove unbound AAV particles and treated with Proteinase K to release AAV genomes for qPCR quantification. For transduction assays, cells were incubated with the AAVs for 24 h at 37 °C and assayed with Britelite plus (Perkin–Elmer, 6066766) following the manufacturer's protocol.

### Interpreting the production fitness and liver-related functional fitness models

We first assessed the effect of each amino acid residue on each predicted function. We fixed each amino acid at each position in the 7-mer one at a time while randomizing the residues at the rest of the positions as in ref. 14. However, we used regression models whereas the referenced study used classification models. Therefore, we used a different importance summarization metric. We used the production

fitness and liver-related functional fitness models to score the functional fitness of the randomized variants ($n = 100K$) to create a distribution in each scenario. The randomized variants were filtered for only production-fit variants using our production fitness model; this was done to eliminate non-production-fit variants that would impact the accuracy of the models trained to only make predictions in the context of production-fit variants (this is a key feature of the Fit4Function approach). We used the mean of the distributions in each case to indicate the overall effect of each fixed residue on each function. The mean predictions for each distribution were z-score normalized to predictions made for randomized 7-mer sequences with no fixed residues at any positions.

Next, we assessed the effect of different physicochemical properties on the predicted functions. For this, we built six surrogate concept models[30] to interpret each of the six functional models. Unlike the original models that are trained on the amino acid sequence of each variant, each concept surrogate model is trained on features representing the physicochemical properties (concepts) of the training variants. To learn how much of those concepts are captured by the original models, the surrogate models are trained to predict the predictions of the original models. In other words, while the original models are trained to learn to predict functional enrichment (fitness) from the sequences of variants, the surrogate models are trained to predict what the original models would predict from the concept features of variants. If the surrogate model can closely learn (replicate) the predictions of the original model by training on the concept features (not the sequences), this means that the original model made predictions largely from those concepts. We built the surrogate concept models as linear regression lasso models, with 10-fold cross-validation for a training set of an arbitrary size of 120K variants, trained on each of twelve physicochemical properties of the 7-mer sequences (i.e., a total of 84 features): the volume of the side chain (normalized), hydropathy (normalized), polarity, hydrogen donor, hydrogen acceptor, positive charge, negative charge, aliphatic, aromatic, sulfur, hydroxyl, and amide. We used the same properties considered in a prior interpretability study[14] but with the assumption that histidine is not positively charged in the pH of the environments relevant to our functions. We used $R^2$ to assess the extent to which the variation of the original model's predictions was explained by those of the surrogate models.

Finally, we assessed the effect of first- and second-degree interactions between residues in the 7-mer on the predicted functions. We built a set of surrogate concept linear regression lasso models with the hot-encoding of first-degree interactions (20 amino acids × 7 positions = 140 features) and another set of models with the hot-encoding of both first- and second-degree interactions (400 combinations of amino acids at two positions × 21 possible dual-position combinations = 8400 features). In each case, data from the 150K Fit4Function variants in the screens of the six functions, excluding non-detected variants, were used for training. $R^2$ was used to assess the extent to which the variation of the original model's predictions was explained by those of the surrogate models.

### Statistics and reproducibility

Microsoft Excel and GraphPad Prism 9 were used for the analysis of experimental data. Python was used for large computational analyses and modeling. The reproducibility of the production fitness scores was tested via 10K variants common to the assessment library and modeling library as shown in Fig. 2d. Biological triplicates were performed for the binding or transduction of different cell lines using the Fit4Function library versus an NNK library and their reproducibility was quantified via pairwise Pearson correlation as shown in Fig. 3c and Supplementary Fig. 5. The Fit4Function library was screened in four mice, and the reproducibility of the biodistribution in eight tissues was assessed in the form of replication quality between pairs of animals as shown in Fig. 3e and Supplementary Fig. 4. The MultiFunction library

was screened across in vitro (three replicates for production fitness, four replicates for binding and transduction) and in vivo ($n = 3$ mice) assays, and the reproducibility within each assay was assessed in the form of replication quality between pairs of replicates as shown in Supplementary Fig. 6. For the comparison between AAV9 and the seven MultiFunction variants, in vitro cell transduction was assessed with four replicates per group (Fig. 4e and Supplementary Fig. 7d; error bars show the standard deviation from the mean) and unpaired, one-sided $t$-tests were conducted on log-transformed values, with Bonferroni correction for multiple hypotheses. The in vivo characterization was performed using five female mice per group except for the no-injection control group (Supplementary Fig. 7b and c, $n = 3$ mice; error bars in Supplementary Fig. 7c show the standard deviation from the mean), experimenters were not blinded to the sample groups, and unpaired, one-sided $t$-tests were conducted on log-transformed values, with Bonferroni correction for multiple hypotheses. Other statistical tests are described in the text. No animals or samples were excluded from the analysis. The variant BI152 was unintentionally excluded from the rhesus macaque study due to a library assembly error. The number of replicates was chosen based on prior data that indicated large effect sizes. ML models were trained in repetition with randomized subsampling and parameterization to ensure reproducibility. Training and testing sample sizes are described in the "Methods" section for each model. Models were tested using independent (blind) datasets where possible, i.e., testing was conducted using an independent assessment library for production fitness (Fig. 2f) and independent measurements in a separate animal (Fig. 3f).

### Reporting summary

Further information on research design is available in the Nature Portfolio Reporting Summary linked to this article.

## Data availability

Data generated in the study that are needed to reproduce, verify, and extend the findings of the study are available in a Zenodo repository under accession code[31] https://doi.org/10.5281/zenodo.8401253. Source data files required for reproducing the manuscript plots are provided in the Zenodo repository https://doi.org/10.5281/zenodo.8388031. as 'Source Data Files.zip'. In addition, NGS data will be made available in the NCBI Sequence Read Archive at the time of publication under BioProject ID: PRJNA1131359. Full plasmid sequences and plasmids for the production of BI151–BI157 are available on Addgene (ID numbers 209523–209529).

## Code availability

All algorithms are described in the main text or "Methods" section. Code and associated data produced in the study that are needed to reproduce, verify, and extend the findings of the study are available in the Zenodo repository under accession code[31] https://doi.org/10.5281/zenodo.8401253. Updated code will be maintained in the GitHub repository: https://github.com/vector-engineering/fit4function.

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

## Acknowledgements
We thank the members of the Deverman laboratory for continuous, extensive discussions of the project; Thouis R. Jones, D. R. Mani, and Andrew Barry for reviewing the manuscript; Sandrine Muller for discussions; and Andrew Steinsapir for assistance in organizing the Fit4-Function library cynomolgus macaque study. This work in the Deverman laboratory was supported by the National Institutes of Health (NIH) Common Fund and the National Institute of Neurological Disorders and Stroke through the Somatic Cell Genome Engineering Consortium (UG3NS111689), a Brain Initiative award funded through the National Institute of Mental Health (UG3MH120096), the Stanley Center for Psychiatric Research, and Apertura Gene Therapy (B.E.D.), a Broad Institute Shark Tank award (F.E.E.), and a Broad Ignite award (Y.A.C.). K.L. was funded by the Novo Nordisk Foundation Center for Genomic Mechanisms of Disease (NNF21SA0072102), the Stanley Center for Psychiatric Research, a National Institute of Mental Health R01 (MH109903) and U01 (MH121499), Simons Foundation Autism Research Initiative awards (515064 and 735604), the Lundbeck Foundation (R223-2016-721 and R350-2020-963), a National Institute of Diabetes and Digestive and Kidney Diseases U01 (DK078616), and Broad Next10 (K.L.). The rhesus macaque study was supported by the NIH Somatic Cell Genome Editing (SCGE) Program including the Nonhuman Primate Testing Center for Evaluation of Somatic Cell Genome Editing Tools (U42 OD027094) and the base operating grant for the California National Primate Research Center (P51-OD011107) awarded to A.F.T.

## Author contributions
This study was conceptualized and supervised by B.E.D. and F.E.E. Its methods were determined by B.E.D., F.E.E., K.Y.C., A.T.C., and Y.A.C. The investigation was conducted by F.E.E., K.Y.C., A.T.C., S.P., I.G.T., Q.H., Q.Z., J.J., C.K., P.P.B., B.Z., M.P., and A.F.T. Formal analysis was performed by F.E.E. and A.T.C. Visualization was done by A.T.C., F.E.E., Y.A.C., and B.E.D. Funding for this study was acquired by B.E.D., F.E.E., Y.A.C., and A.F.T. The original draft was written by F.E.E., Y.A.C., B.E.D., and A.T.C., reviewed and edited by F.E.E., Y.A.C., B.E.D., A.T.C., and A.F.T., and approved by all co-authors.

## Competing interests
BED is a scientific founder at Apertura Gene Therapy and a scientific advisory board member at Tevard Biosciences. BED, FEE, and KYC are named inventors on patent applications filed by the Broad Institute of MIT and Harvard related to the design and use of Fit4Function libraries (WO2021222636) and AAV sequences developed as part of this study. The remaining authors declare that they have no competing interests.
