## [Peer Review File · Nature Communications]

Reviewers' Comments:

Reviewer #1:

Remarks to the Author:

The authors have done a good job addressing the comments from the previous round of reviews and have significantly improved the manuscript.

Reviewer #2:

Remarks to the Author:

In this revised manuscript, Eid et al adequately delineate key differences between their Fit4Function approach and other published efforts to bridge ML with AAV engineering. The authors should be commended for carrying out NHP studies and demonstrating encouraging results with several new AAV variants that display improved liver transduction compared to AAV9. Several arguments made by the authors with regard to "more efficient screening..." "co-optimization of multiple traits..." are supported by the data. Nonetheless, the major (and common) criticisms raised by each of the reviewers with regard to significance, novelty and new insight are largely unaddressed. Notably, the following concerns remain:

1. While technically rigorous, the methodology described herein has not been rigorously tested against a challenging problem in the AAV gene therapy space. For instance, the peptide insert approach on AAV9 capsids has been utilized successfully by several groups and it appears that this strategy does not overwhelmingly affect traits such as viral titer with engineered leads. Therefore, the authors claim that this approach leads to selection of capsids with improved titers is overstated. Further, if multiple traits were indeed impacted by the peptide insert, the data does not provide sufficient evidence that the ML approach led to selective liver tropism (more likely improved uptake in general as evidenced in vitro).

2. As alluded to by multiple reviewers, the liver has multiple AAV candidates in clinical trials (AAV5, LK03, AAV8, AAV.rh10, AAVS3 etc) and as such does not appear to be a subject of critical need. Notably, as indicated in the earlier review, AAV9 is more of an exception for liver GT applications - hence, the application of this method to improve AAV9 for liver applications does not particularly advance the field. Moreover, it is unclear whether any of the new variants reported herein offer any significant advantage over capsids listed above.

3. As the authors recognize, improved gene transfer efficiency in the liver does not necessarily imply the ability to lower dose (this is target specific - number of hepatocytes corrected vs amount of transgene product desired at a particular dose). Hence the implications of the newly included NHP liver data are unclear. Notably, the authors claim increased transduction as represented by enrichment - is this data based on normalized vector genome (vg) copy number or transcript levels (not evident from figure, legend or discussion)? If vg, that would simply imply increased liver uptake, which typically mirrors in vitro cellular uptake traits. Hence, attributing the observed improvements to ML guided improvements in liver tropism is misleading.

4. In summary, this is a compelling dataset that supports an improved and possibly more streamlined, time/resource-effective screening method for engineering AAV capsids. However, the choice of the specific application in improving liver gene transfer is underwhelming, the prospective applications of engineered leads for liver GT and the lack of significant new insight on how these multiple traits (titer vs in vitro transduction vs mouse vs NHP) are connected by the enriched sequences - all of which remain significant weaknesses of the overall study.

Reviewer #4:

Remarks to the Author:

AAV capsid design is vital for efficient gene delivery. Eid et al developed a machine-learning approach to the systematical design of AAV capsid. In this work, the authors trained LSTM models to predict the functions of 7-mer amino acids (AA) inserted into VP1 residues 588-589 of AAV9.

The sequences are sampled from the amino acids space rather than the nucleotide space. The result shows less bias caused by codons in this way. The prediction result is accurate and highly correlated with the measurement. Most training and test process of the machine learning model meets the specification.

I have some questions about the detail of the training and testing process. These details should be declared for the reader to better understand the processes.

Major:

In this study, the authors trained multiple models to predict the fitness functions of 7-mer amino acids (AAs), which showed strong performance, suggesting that the models have learned the underlying sequence rules. For instance, specific k-mers or AA interactions might significantly impact high production or other functions. Extracting these rules from the models could offer valuable biological insights to readers. Nonetheless, further analysis of all the models is lacking. Therefore, I strongly recommend that the authors employ techniques such as counting k-mer frequency or using neural network interpretation methods, such as Saliency Maps or AA perturbation, to identify rules that contribute to different functions.

Minor:

1. Fit4Function enables reproducible data. But it's not clear to me why the Fit4Function library consistently yield higher replication quality than the NNK library? Could the author further explain this?

2. In line 382, the authors discussed: "variants that are maximally optimized for multiple objectives may not exist, especially in cases where performance on functions are negatively correlated." I'd like to know based on the models in this paper, which groups of objectives are negatively correlated. For example, could the author provide a spearman correlation matrix to visualize the associations and correlations among different experiments read out, which can clearly show the positive and negative correlated groups of objects.

Point-by-point response to reviewer comments

Reviewer #1 (Remarks to the Author):

The authors have done a good job addressing the comments from the previous round of reviews and have significantly improved the manuscript.

We thank the reviewer for their initial feedback and are glad that our revisions have addressed their comments.

Reviewer #2 (Remarks to the Author):

In this revised manuscript, Eid et al adequately delineate key differences between their Fit4Function approach and other published efforts to bridge ML with AAV engineering. The authors should be commended for carrying out NHP studies and demonstrating encouraging results with several new AAV variants that display improved liver transduction compared to AAV9. Several arguments made by the authors with regard to “more efficient screening...” “co-optimization of multiple traits...” are supported by the data. Nonetheless, the major (and common) criticisms raised by each of the reviewers with regard to significance, novelty and new insight are largely unaddressed. Notably, the following concerns remain:

1. While technically rigorous, the methodology described herein has not been rigorously tested against a challenging problem in the AAV gene therapy space. For instance, the peptide insert approach on AAV9 capsids has been utilized successfully by several groups and it appears that this strategy does not overwhelmingly affect traits such as viral titer with engineered leads. Therefore, the authors claim that this approach leads to selection of capsids with improved titers is overstated.

The reviewer is correct that the use of 7-mer peptides to redirect the tropism of AAV9 (and other AAV capsids) is a well-established approach. However, as assessed and described in this manuscript in an unbiased library format (Fig. 2b), the majority of 7-mer insertions are not compatible with production that is on par with the parental capsid AAV9. Here, for the first time, we demonstrate that the production fitness of these capsids can be predicted with high accuracy.

We have not made the claim in the manuscript that this approach leads to the selection of capsids with improved titers. Because this point may also be confusing to readers, we modified our explanation in the text: “Note that the “production-fit” distribution includes variants that produce better than, as well as, or less well than AAV9; “production-fit” is not defined as having a production fitness score greater than that of AAV9.”

This confusion may have been caused because we identified a “high production fitness” distribution (Fig. 2b) and used this phrase to describe all variants included in Fit4Function libraries. We have now made modifications throughout the text to differentiate the two populations as production-fit versus non-fit, as opposed to high fitness versus low fitness. This should help clear up the confusion that may stem from an assumption that variants within the “high production fitness” distribution are predicted to out produce AAV9. Our AAV9 fitness data

demonstrates that its measured production fitness falls within the high production fitness distribution.

Further, if multiple traits were indeed impacted by the peptide insert, the data does not provide sufficient evidence that the ML approach led to selective liver tropism (more likely improved uptake in general as evidenced in vitro).

We agree with the reviewer that the multi-trait liver-targeting capsids may not have led to capsids with selective liver tropism. We have not made the claim that this effort has identified capsids with selective liver tropism, nor was this our objective. Our claim is that our approach led to the development of capsids with greater liver enrichment/efficient liver transduction across species. Our data is not consistent with a general improved uptake phenotype because six of the seven individually tested multifunction variants exhibited reduced targeting of the brain, spinal cord, and kidney compared to AAV9 (Fig. 4d).

2. As alluded to by multiple reviewers, the liver has multiple AAV candidates in clinical trials (AAV5, LK03, AAV8, AAV.rh10, AAVS3 etc) and as such does not appear to be a subject of critical need.

We can see where reviewer 2 is coming from but we disagree with the points raised in this criticism. First, the Fit4Function methodology addresses a challenging problem in the AAV engineering space. As we describe in the paper, the peptide insertion approach has been utilized to some success by several groups (including ours), i.e., it is possible to select for capsids with a single enhanced function such as crossing the BBB in mice or NHPs, or more efficient transduction of human cell xenograft model. However, it remains incredibly inefficient in the pursuit of capsids with multiple, cross-species traits of interest, e.g., a capsid that can efficiently target a specific organ across mice, NHPs, and human cells. Dozens of selection experiments involving hundreds of animals have been conducted across groups; yet these efforts have only led to the discovery of a couple of examples of capsid families that exhibit more efficient transduction (e.g., of the CNS or muscle) across rodents and primates.

We sought to publish the Fit4Function approach, which represents a significant departure from these conventional in vivo selection approaches, as soon as we verified that it could produce reproducible, ML-compatible data to train sequence-to-function models that can be leveraged in combination to accurately predict multi-trait capsids. We are building on this initial study and will be generating new data and training models aimed at addressing other multi-trait and cross-species gene delivery challenges. The design-build-test cycle, which includes NHP and human cell validation studies, will take at least another year to complete and cannot be included in this publication.

Notably, as indicated in the earlier review, AAV9 is more of an exception for liver GT applications - hence, the application of this method to improve AAV9 for liver applications does not particularly advance the field. Moreover, it is unclear whether any of the new variants reported herein offer any significant advantage over capsids listed above.

In a side-by-side comparison with other AAV serotypes, AAV9 can achieve strong liver transduction in mice (see our data in Extended Data Figure 7, and Figure 4 in <https://www.sciencedirect.com/science/article/pii/S1525001616317324>) and in NHP (new work at the California National Primate Research Center, recently presented at the 2023 Annual ASGCT meeting, Abstract 78: AAV Serotype Tropism and Editing in Young Rhesus Monkeys, Alice F. Tarantal et al., showed that AAV9 transduced several macaque organs including the liver more efficiently than other serotypes including AAV5 and AAV8 <https://annualmeeting.asgct.org/abstracts/abstract-details?abstractId=14987>). We have now added the following to our manuscript: “Notably, AAV9 has a strong liver transduction profile in animals compared to other AAV serotypes, including AAV5 and AAV8¹⁷.” As we wrote in our manuscript, “Liver-directed therapies should benefit from the development of more potent AAV vectors that can be administered at lower doses to reduce the exposure to capsid antigens.” Other groups interested in the liver might decide to apply Fit4Function to their capsid of choice.

3. As the authors recognize, improved gene transfer efficiency in the liver does not necessarily imply the ability to lower dose (this is target specific - number of hepatocytes corrected vs amount of transgene product desired at a particular dose). Hence the implications of the newly included NHP liver data are unclear.

The dosage chosen for a specific indication is certainly indication and transgene specific. However, we are not aware of data that supports the possibility that increasing transduction efficiency would only lead to increased vector genomes per cell without an increase in the number of cells that are transduced. We anticipate that an increase in gene transfer efficiency will enable doses to be lowered for both secreted products and cell autonomous gene products.

Notably, the authors claim increased transduction as represented by enrichment - is this data based on normalized vector genome (vg) copy number or transcript levels (not evident from figure, legend or discussion)? If vg, that would simply imply increased liver uptake, which typically mirrors in vitro cellular uptake traits. Hence, attributing the observed improvements to ML guided improvements in liver tropism is misleading.

We performed two NHP studies - one for biodistribution (vector genomes measured) and one for transduction (transcript levels measured). The details of each study protocol are in the Methods section. We have added wording to the main text and the caption of Figure 4 to make the distinction between the two experiments clearer. The captions for Fig 4f and 4h have been edited as follows (underlined): “A 100K variant Fit4Function library was injected intravenously into a cynomolgus macaque and the distribution of vector genomes was assessed four hours later.” and “The macaque liver transduction efficiency, measured by transcript levels 4-weeks post-administration, for the seven individually characterized liver MultiFunction variants are shown (n = 2 rhesus macaques).”

Therefore, our statements about improvements to liver tropism are not solely based on vector genome accumulation and are not in any way misleading. Indeed, we found that all seven of the

individually characterized capsids chosen from our mouse biodistribution (vg readout) and human cell binding and transduction MultiFunction library data were validated by (1) increased copy number in mouse, cynomolgus macaque, and human cells, (2) increased liver gene expression (as measured by mRNA transcript enrichment) in mice, rhesus macaque, and human cells, and (3) individually at the reporter protein level in mice and human cells (Extended Fig. 7 and 8).

4. In summary, this is a compelling dataset that supports an improved and possibly more streamlined, time/resource-effective screening method for engineering AAV capsids. However, the choice of the specific application in improving liver gene transfer is underwhelming, the prospective applications of engineered leads for liver GT and the lack of significant new insight on how these multiple traits (titer vs in vitro transduction vs mouse vs NHP) are connected by the enriched sequences - all of which remain significant weaknesses of the overall study.

We respectfully disagree that what we have achieved with this liver focused MultiFunction effort is underwhelming. We engineered capsids that transduced the macaque liver and multiple human hepatocyte-like cell lines significantly better than AAV9, a capsid that has relatively strong liver targeting compared to other serotypes used in liver gene therapies.

While there are other capsids that are capable of transducing liver cells in humans as highlighted by two recently approved liver-directed gene therapies, the doses used for each of these therapies (Hemgenix, 2×10^{13} vg/kg; Roctavian, 6×10^{13} vg/kg) remain high. As with any technology, new iterations will improve the product. Capsids engineered to provide more efficient liver delivery should enable efficient expression at lower doses, which would reduce cost of goods and enable lower costs to patients and the health care system (Hemgenix and Roctavian were launched at \$3.5M/dose and ~1.5M Euros/dose, respectively). Lower doses may also allow for shorter courses of immunosuppressive regimens.

As it stands, our strategy has achieved a significant advance in the field of biological ML and is an essential step towards using ML to search the vast sequence space of modified capsids (not limited to single peptide insertion libraries) for variants possessing multiple traits across different species. No other group has reported a ML-guided approach with the level of accuracy and generalizability demonstrated in our work, not to mention applying such an approach to accurately predict capsids that have multiple functions. Our strategy enables us to systematically determine the functional assays or combinations thereof that drive cross-species transferability. This is again a new advance that no other group has achieved. We are currently applying Fit4Function to other types of modified capsids, targeting other disease-relevant cell types and/or receptors expressed on these cells. That next step will require a large screening effort in multiple species, including in NHP, and will be published in a future manuscript.

Reviewer #4 (Remarks to the Author):

AAV capsid design is vital for efficient gene delivery. Eid et al developed a machine-learning approach to the systematical design of AAV capsid. In this work, the authors trained LSTM models to predict the functions of 7-mer amino acids (AA) inserted into VP1 residues 588-589 of

AAV9. The sequences are sampled from the amino acids space rather than the nucleotide space. The result shows less bias caused by codons in this way. The prediction result is accurate and highly correlated with the measurement. Most training and test process of the machine learning model meets the specification.

I have some questions about the detail of the training and testing process. These details should be declared for the reader to better understand the processes.

Major:

In this study, the authors trained multiple models to predict the fitness functions of 7-mer amino acids (AAs), which showed strong performance, suggesting that the models have learned the underlying sequence rules. For instance, specific k-mers or AA interactions might significantly impact high production or other functions. Extracting these rules from the models could offer valuable biological insights to readers. Nonetheless, further analysis of all the models is lacking. Therefore, I strongly recommend that the authors employ techniques such as counting k-mer frequency or using neural network interpretation methods, such as Saliency Maps or AA perturbation, to identify rules that contribute to different functions.

We agree with the reviewer that investigating the model behavior can lead to biological insights. We conducted an interpretation study to understand what signals each model picked up. We have summarized this new analysis in the Results section and Extended Data Figure 9, and added to the Methods section. This is the text added to the Results section:

“Our finding that six of seven individually tested MultiFunction liver-enriched variants exhibited reduced targeting of the brain, spinal cord, and kidney compared to AAV9 (Fig. 4d) suggested that the models did not simply learn to predict capsid variants with generally improved uptake into cells. To gain insight into the features learned by the models, we investigated how each amino acid residue at each position in the 7-mer, different physicochemical properties at each position, as well as epistatic interactions contribute to explaining the signals learned by each of the models. Across the six models, we observed that certain charged residues had a large overall impact on the predicted scores regardless of which position they occupied within the 7-mer; the presence of Cysteine or Tryptophan at any position led to reduced production fitness predicted scores, and the presence of Arginine, Lysine, or Cysteine tended to lead to predictions of increased liver cell targeting (Extended Data Fig. 9a). Positive and negative charges contributed to increases and decreases in predicted scores, respectively (Extended Data Fig. 9b). However, physicochemical properties alone could not fully account for the signal learned by the models (Extended Data Fig. 9b). The models learned both first-degree (single residue) and second-degree (paired residues) interactions but those interactions could not fully explain the original model's predictions, indicating that more complex signals or higher order epistasis drive the functional fitness (Extended Data Fig. 9c).”

Minor:

1. Fit4Function enables reproducible data. But it's not clear to me why the Fit4Function library consistently yield higher replication quality than the NNK library? Could the author further explain this?

Fit4Function libraries are designed to have a known and limited membership and to have low bias at the AA level. In contrast, with NNK libraries, the likelihood of generating quantitative data for each variant is reduced due to AA bias (Fig. 2c) and the much larger number of variants in the library. This affects replication quality because it is both more challenging to obtain accurate measurements of the amount of each variant in the unselected NNK virus library and after selection from the functional assays, which reduces the reliability of the calculated enrichment scores.

Our manuscript has the following text: “Fit4Function libraries are designed to enable the generation of reproducible and ML-compatible functional screening data. Specifically, the library is limited to a moderate size that enables deeper sequencing depth and samples only production-fit variants, which both enable more quantitative and reliable detection of each variant in the library.”

We have added the following new text to the manuscript to clarify this point:

These results are unsurprising because NNK libraries have theoretically millions of random variants, some of which can be at levels that are two or more orders of magnitude higher than other variants in the library. As a result, state-of-art sequencing is not able to accurately quantify a great number of the underrepresented variants in NNK libraries before and after screens. In addition, the unknown variant membership in NNK libraries and the resulting low quality data mean that ML models cannot be trained to discriminate between truly non-functional variants and variants that cannot be reliably detected. In comparison, the Fit4Function library sizes (we set these at 240K or smaller) have a manageable, known membership and focus on production-fit variants, which increases the likelihood of detection for each variant. Critically, this allows for learning from both positive and negative data. Furthermore, Fit4Function libraries used for training functional models are populated with variants sampled uniformly across the sequence space. Compared to NNK libraries, the higher replication quality and reduced bias in Fit4Function libraries enable more accurate modeling.

2. In line 382, the authors discussed: “variants that are maximally optimized for multiple objectives may not exist, especially in cases where performance on functions are negatively correlated.” I’d like to know based on the models in this paper, which groups of objectives are negatively correlated. For example, could the author provide a spearman correlation matrix to visualize the associations and correlations among different experiments read out, which can clearly show the positive and negative correlated groups of objects.

We have now added a reference to the Extended Data Fig. 4 in the discussion (line 382, previously); Extended Data Fig. 4 shows a correlation matrix including examples of anti-correlated tropisms.

Reviewers' Comments:

Reviewer #2:

Remarks to the Author:

Concerns addressed

Reviewer #4:

Remarks to the Author:

The authors have addressed my comments properly.